# LOG-DENSENET: HOW TO SPARSIFY A DENSENET

## ABSTRACT

Skip connections are increasingly utilized by deep neural networks to improve accuracy and cost-efficiency. In particular, the recent DenseNet is efficient in computation and parameters, and achieves state-of-the-art predictions by directly connecting each feature layer to all previous ones. However, DenseNet's extreme connectivity pattern may hinder its scalability to high depths, and in applications like fully convolutional networks, full DenseNet connections are prohibitively expensive. This work first experimentally shows that one key advantage of skip connections is to have short distances among feature layers during backpropagation. Specifically, using a fixed number of skip connections, the connection patterns with shorter backpropagation distance among layers have more accurate predictions. Following this insight, we propose a connection template, Log-DenseNet, which, in comparison to DenseNet, only slightly increases the backpropagation distances among layers from 1 to $(1+\log_2 L)$, but uses only $L \log_2 L$ total connections instead of $O(L^2)$. Hence, Log-DenseNets are easier to scale than DenseNets, and no longer require careful GPU memory management. We demonstrate the effectiveness of our design principle through ablation studies and by showing better performance than DenseNets on *tabula rasa* semantic segmentation, and competitive results on visual recognition.

## 1 INTRODUCTION

Deep neural networks have been improving performance for many machine learning tasks, scaling from networks like AlexNet (Krizhevsky et al., 2012) to increasingly more complex and expensive networks, like VGG (Simonyan & Zisserman, 2014), ResNet (He et al., 2016) and Inception (Christian Szegedy & Alemi, 2017). Continued hardware and software advances will enable us to build deeper neural networks, which have higher representation power than shallower ones. However, the payoff from increasing the depth of the networks only holds in practice if the networks can be trained effectively. It has been shown that naïvely scaling up the depth of networks actually decreases the performance (He et al., 2016), partially because of vanishing/exploding gradients in very deep networks. Furthermore, in certain tasks such as semantic segmentation, it is common to take a pre-trained network and fine-tune, because training from scratch is difficult in terms of both computational cost and reaching good solutions. Overcoming the vanishing gradient problem and being able to train from scratch are two active areas of research.

Recent works attempt to overcome these training difficulties in deeper networks by introducing skip, or shortcut, connections (Long et al., 2015; Hariharan et al., 2015; Srivastava et al., 2015; He et al., 2016; Larsson et al., 2017; Huang et al., 2017) so the gradient reaches earlier layers and compositions of features at varying depth can be combined for better performance. In particular, DenseNet (Huang et al., 2017) is the extreme example of this, concatenating all previous layers to form the input of each layer, i.e., connecting each layer to all previous ones. However, this incurs an $O(L^2)$ run-time complexity for a depth $L$ network, and may hinder the scaling of networks. Specifically, in fully convolutional networks (FCNs), where the final feature maps have high resolution so that full DenseNet connections are prohibitively expensive, Jégou et al. (2017) propose to cut most of connections from the mid-depth. To combat the scaling issue, Huang et al. (2017) propose to halve the total channel size a number of times. Futhermore, Liu et al. (2017) cut 40% of the channels in DenseNets while maintaining the accuracy, suggesting that much of the $O(L^2)$ computation is redundant. Therefore, it is both necessary and natural to consider a more efficient design principle for placing shortcut connections in deep neural networks.

In this work, we address the scaling issue of skip connections by answering the question: if we can only afford the computation of a limited number of skip connections and we believe the network needs to have at least a certain depth, where should the skip connections be placed? We design experiments to show that with the same number of skip connections at each layer, the networks can have drastically different performance based on where the skip connections are. In particular, we summarize this result as the following design principle, which we formalize in Sec. 3.2: **given a fixed number of shortcut connections to each feature layer, we should choose these shortcut connections to minimize the distance among layers during backpropagation.**

Following this principle, we design a network template, Log-DenseNet. In comparison to DenseNets at depth $L$, Log-DenseNets cost only $L \log L$, instead of $O(L^2)$ run-time complexity. Furthermore, Log-DenseNets only slightly increase the short distances among layers during backpropagation from 1 to $1 + \log L$. Hence, Log-DenseNets can scale to deeper and wider networks, even without custom GPU memory managements that DenseNets require. In particular, we show that Log-DenseNets outperform DenseNets on *tabula rasa* semantic segmentation on CamVid (Brostow et al., 2008), while using only half of the parameters, and similar computation. Log-DenseNets also achieve comparable performance to DenseNet with the same computations on visual recognition data-sets, including ILSVRC2012 (Russakovsky et al., 2015). In short, our contributions are as follows:

- We experimentally support the design principle that with a fixed number of skip connections per layer, we should place them to minimize the distance among layers during backpropagation.

- The proposed Log-DenseNets achieve small $1 + \log_2 L$ between-layer distances using few connections ($L \log_2 L$), and hence, are scalable for deep networks and applications like FCNs.

- The proposed network outperforms DenseNet on CamVid for *tabula rasa* semantic segmentation, and achieves comparable performance on ILSVRC2012 for recognition.

## 2  BACKGROUND AND RELATED WORKS

**Skip connections.** The most popular approach to creating shortcuts is to directly add features from different layers together, with or without weights. Residual and Highway Networks (He et al., 2016; Srivastava et al., 2015) propose to sum the new feature map at each depth with the ones from skip connections, so that new features can be understood as fitting residual features of the earlier ones. FractalNet (Larsson et al., 2017) explicitly constructs shortcut networks recursively and averages the outputs from the shortcuts. Such structures prevent deep networks from degrading from the shallow shortcuts via "teacher-student" effects. (Huang et al., 2016) implicitly constructs skip connections by allowing entire layers to be dropout during training. DualPathNet (Chen et al., 2017) combines the insights of DenseNet (Huang et al., 2017) and ResNet (He et al., 2016), and utilizes both concatenation and summation of previous features.

**Run-time Complexity and Memory of DenseNets.** DenseNet (Huang et al., 2017) emphasizes skip connections by directly connecting each layer to all previous layers. However, this quadratic complexity may prevent DenseNet from scale to deep and wide models. In order to scale, DenseNet applies block compression, which halves the number of channels in the concatenation of previous layers. DenseNet also opts not to double the output channel size of conv layers after downsampling, which divides the computational cost of each skip connection. These design choices enable DenseNets to be deep for image classification where final layers have low resolutions. However, final layers in FCNs for semantic segmentation have higher resolution than in classification. Hence, to fit models in the limited GPU memory, FC-DenseNets (Jégou et al., 2017) have to cut most of their skip connections from mid-depth layers.

Furthermore, a naïve implementation of DenseNet requires $O(L^2)$ memory, because the inputs of the $L$ convolutions are individually stored. Though there exist $O(L)$ implementations via memory sharing among layers (Liu, 2017), they require custom GPU memory management, which is not supported in many existing packages. Hence, one may have to use custom implementations and re-compile packages for memory efficient Densenets, e.g., it costs a thousand lines of C++ on Caffe (Li, 2016). Our work recognizes the contributions of DenseNet's architecture to utilize skip connections, and advocates for the efficient use of compositional skip connections to shorten the distances among feature layers during backpropagation. Our design principle can especially help applications like FC-DenseNet (Jégou et al., 2017) where the network is desired to be at least a certain depth, but only a limited number of shortcut connections can be formed.

**Network Compression.** A wide array of works have proposed methods to compress networks by reducing redundancy and computational costs. (Denton et al., 2014; Kim et al., 2016; Ioannou et al., 2016) decompose the computation of convolutions at spatial and channel levels to reduce convolution complexity. (Hinton et al., 2014; Ba & Caruana, 2014; Romero et al., 2015) propose to train networks with smaller costs to mimic expensive ones. (Liu et al., 2017) uses $L1$ regularization to cut 40% of channels in DenseNet without losing accuracy. These methods, however, cannot help in applications that cannot fit the complex networks in GPUs in the first place. This work, instead of cutting connections arbitrarily or post-design, advocates a network design principle to place skip connections intelligently to minimize between-layer distances.

## 3 FROM DENSENET TO LOG-DENSENET

### 3.1 PRELIMINARY ON DENSENETS

Formally, we call the feature layers in a feed-forward convolutional network as $x_0, x_1, ..., x_L$, where $x_0$ is the result of the initial convolution on the input image. Each $x_i$ is a transformation $f_i$ with parameter $\theta_i$ and takes input from a subset of $x_0, ..., x_{i-1}$. E.g., a traditional feed-forward network has $x_i = f_i(x_{i-1}; \theta_i)$, and the recent DenseNet (Huang et al., 2017) proposes to form each feature layer $x_i$ using all previous features layers, i.e.,

$$x_i = f_i(\text{concat}(\{x_j : j = 0, ..., i - 1\}) ; \theta_i), \tag{1}$$

where concat($\bullet$) concatenates all features in its input collection along the feature channel dimension. Each $f_i$ is a bottleneck structure (Huang et al., 2017), i.e., BN-ReLU-1x1conv-BN-ReLU-3x3conv, where the final conv produces the growth rate $g$ number of channels, and the bottleneck 1x1 conv produces $4g$ channels of features. DenseNet also organizes layers into $n_{block}$ number of blocks. Between two contiguous blocks, there is a block compression using a 1x1conv-BN-ReLU, followed by an average pooling, to downsample previous features for deeper and coarser layers. In practice, $n_{block}$ is small in visual recognition architectures (He et al., 2016; Christian Szegedy & Alemi, 2017; Huang et al., 2017).

The direct connections among layers in DenseNet are believed to introduce implicit deep supervision (Lee et al., 2015) in intermediate layers, and reduce the vanishing/exploding gradient problem by enabling direct influence between any two feature layers. Inspired by this belief, we propose a design principle to organize the skip connections: with a fixed connection budget, we should minimize the connection distance among layers.

### 3.2 MAXIMUM BACKPROPAGATION DISTANCE

To formalize our design principle, we consider each $x_i$ as a node in a graph, and the directed edge $(x_i, x_j)$ exists if $x_i$ takes direct input from $x_j$. The *backpropagation distance* (BD) from $x_i$ to $x_j$ $(i > j)$ is then the length of the shortest path from $x_i$ to $x_j$ on the graph. Then we define the *maximum backpropagation distance* (MBD) as the maximum BD among all pairs $i > j$. Then DenseNet has a MBD of 1, if we disregard transition layers, but at the cost of $O(L^2)$ connections. We next propose short connection patterns for when the connection budget is $O(L \log L)$. In comparison to DenseNet, The proposed Log-DenseNet increases the MBD to $1 + \log_2 L$ while using only $O(L \log L)$ connections. Since the current practical networks have less than 1000 depths, the proposed method has a single-digit MBD.

### 3.3 LOG-DENSENET

For simplicity, we let $\log(\bullet)$ denote $\log_2(\bullet)$. In a proposed Log-Dense Network, each layer $i$ takes direct input from at most $\log(i) + 1$ number of previous layers, and these input layers are exponentially apart from depth $i$ with base 2, i.e.,

$$x_i = f_i(\text{concat}(\{x_{i-\lfloor 2^k \rceil} : k = 0, ..., \lfloor \log(i) \rfloor\}) ; \theta_i), \tag{2}$$

where $\lfloor \bullet \rceil$ is the nearest integer function and $\lfloor \bullet \rfloor$ is the floor function. For example, the input features for layer $i$ are layer $i - 1, i - 2, i - 4, ...$. We define the input index set at layer $i$ to be $\{i - \lfloor 2^k \rceil : k = 0, ..., \lfloor \log(i) \rfloor\}$. We illustrate the connection in Fig. 1b. Since the complexity of layer $i$ is $\log(i) + 1$, the overall complexity of a Log-DenseNet is $\sum_{i=1}^{L}(\log(i) + 1) \leq L + L \log L = \Theta(L \log L)$, which is significantly smaller than the quadratic complexity, $\Theta(L^2)$, of a DenseNet.

**Log-DenseNet V1: independent transition.** Following Huang et al. (2017), we organize layers into blocks. Layers in the same block have the same resolution; the feature map side is halved after

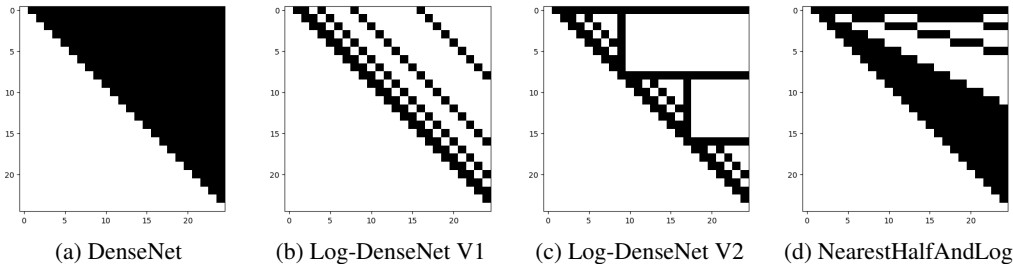

| (a) DenseNet | (b) Log-DenseNet V1 | (c) Log-DenseNet V2 | (d) NearestHalfAndLog |

Figure 1: Connection illustration for $L = 24$. Layer 0 is the initial convolution. $(i, j)$ is black means $x_j$ takes input from $x_i$; it is white if otherwise. We assume there is a block transition at depth 12 for Log-DenseNet V2. **(d)** illustrates NearestHalfAndLog, a connection pattern with $O(L^2)$ connection budget for experiments in Sec. 4.1; it greatly improves over NearestHalf.

each block. In between two consecutive blocks, a transition layer will shrink all previous layers so that future layers can use them in Eq 2. We define a *pooling transition* as a 1x1 conv followed by a 2x2 average pooling, where the output channel size of the conv is the same as the input one. We refer to $x_i$ after $t$ number of pooling transition as $x_i^{(t)}$. In particular, $x_i^{(0)} = x_i$. Then at each transition layer, for each $x_i$, we find the latest $x_i^{(t)}$, i.e., $t = max\{s \geq 0 : x_i^{(s)} \text{exists}\}$, and compute $x_i^{(t+1)}$. We abuse the notation $x_i$ when it is used as an input of a feature layer to mean the appropriate $x_i^{(t)}$ so that the output and input resolutions match. Unlike DenseNet, we independently process each early layer instead of using a pooling transition on the concatenated early features, because the latter option results in $O(L^2)$ complexity per transition layer, if at least $O(L)$ layers are to be processed. Since Log-DenseNet costs $O(L)$ computation for each transition, the total transition cost is $O(L \log L)$ as long as we have $O(\log L)$ transitions.

**Log-DenseNet V2: block compression.** Unfortunately, many neural network packages, such as TensorFlow, cannot compute the $O(L)$ 1x1 conv for transition efficiently: in practice, this $O(L)$ operation costs about the same wall-clock time as the $O(L^2)$-cost 1x1 conv on the concatenation of the $O(L)$ layers. To speed up transition and to further reduce MBD, we propose a block compression for Log-DenseNet similar to the block compression in DenseNet (Huang et al., 2017). At each transition, the newly finished block of feature layers are concatenated and compressed into $g \log L$ channels using 1x1 conv. The other previous compressed features are concatenated, followed by a 1x1 conv that keep the number of channels unchanged. These two blocks of compressed features then go through 2x2 average pooling to downsample, and are then concatenated together. Fig. 1c illustrates how the compressed features are used when $n_{block} = 3$, where $x_0$, the initial conv layer of channel size $2g$, is considered the initial compressed block. The total connections and run-time complexity are still $O(L \log L)$, at any depth the total channel from the compressed feature is at most $(n_{block} - 1)g \log L + 2g$, and we assume $n_{block} \leq 4$ is a constant. Furthermore, these transitions cost $O(L \log L)$ connections and computation in total, since compressing of the latest block costs $O(L \log L)$ and transforming the older blocks costs $O(\log^2 L)$.

**Log-DenseNet MBD.** The reduction in complexity from $O(L^2)$ in DenseNet to $O(L \log L)$ in Log-DenseNet only increases the MBD among layers to $1 + \log L$. This result is summarized as follows.

**Proposition 3.1.** *For any two feature layers $x_i \neq x_j$ in Log-DenseNet that has $n_{block}$ number of blocks, the maximum backpropagation distance between $x_i$ and $x_j$ is at most $\log |j - i| + n_{block}$.*

This proposition argues that if we ignore pooling layers, or in the case of Log-DenseNet V1, consider the transition layers as part of each feature layer, then any two layers $x_i$ and $x_j$ are only $\log |j-i|+1$ away from each other during backpropagation, so that layers can still easily affect each other to fit the training signals. Sec. 4.1 experimentally shows that with the same amount the connections, the connection strategy with smaller MBD leads to better accuracy. We defer the proof to the appendix. In comparison to Log-DenseNet V1, V2 reduces the BD between any two layers from different blocks to be at most $n_{block}$, where the shortest paths go through the compressed blocks.

**Deep supervision.** Since we cut the majority of the connections in DenseNet when forming Log-DenseNet, we found that having additional training signals at the intermediate layers using deep

| (n,g) | LD,N,E GFLOPS | LD2 GFLOPS | CIFAR10 | | | | CIFAR100 | | | | SVHN | | | |
|---|---|---|---|---|---|---|---|---|---|---|---|---|---|---|
| | | | LD | N | E | LD2 | LD | N | E | LD2 | LD | N | E | LD2 |
| (12,16) | 0.29 | 0.46 | 7.23 | 7.59 | 7.45 | 5.52 | 29.14 | 30.59 | 30.72 | 25.10 | 2.03 | 2.11 | 2.27 | 1.77 |
| (12,24) | 0.66 | 1.03 | 5.98 | 6.46 | 6.56 | 4.88 | 26.36 | 26.96 | 27.80 | 22.50 | 1.94 | 2.10 | 2.05 | 1.63 |
| (12,32) | 1.17 | 1.82 | 5.48 | 6.00 | 6.15 | 4.63 | 24.21 | 24.70 | 25.57 | 21.67 | 1.85 | 1.92 | 1.90 | 1.56 |
| (32,16) | 0.99 | 1.41 | 5.96 | 6.45 | 6.21 | 4.70 | 25.32 | 27.48 | 26.81 | 22.15 | 1.97 | 1.94 | 1.96 | 1.68 |
| (32,24) | 2.23 | 3.16 | 5.03 | 5.74 | 5.43 | 4.21 | 22.73 | 25.08 | 24.80 | 19.58 | 1.77 | 1.82 | 1.95 | 1.62 |
| (32,32) | 3.96 | 5.92 | 4.81 | 5.65 | 4.94 | 4.16 | 21.77 | 23.79 | 23.87 | 19.54 | 1.76 | 1.82 | 1.95 | 1.61 |
| (52,16) | 1.81 | 2.39 | 5.13 | 6.80 | 6.09 | 4.20 | 23.45 | 27.99 | 26.58 | 21.73 | 1.66 | 1.98 | 1.85 | 1.67 |
| (52,24) | 4.07 | 5.35 | 4.34 | 5.83 | 5.03 | 4.04 | 20.99 | 26.07 | 24.19 | 20.13 | 1.64 | 1.90 | 1.80 | 1.69 |
| (52,32) | 7.23 | 9.51 | 4.56 | 6.10 | 4.98 | 4.10 | 20.58 | 24.79 | 23.10 | 19.45 | 1.72 | 1.89 | 1.78 | 1.60 |

Table 1: Performance of various connection patterns with $O(L \log L)$ total connections. Log-DenseNet V1(LD), NEAREST (N) and EVENLY-SPACED (E) have $\log i$ shortcut connections to layer $i$, and have MBD of $1 + \log L$, $\frac{L}{\log L}$ and $\frac{L}{\log L}$. LD clearly outperforms N and E thanks to its low MBD. Log-DenseNet V2 (LD2) outperforms the others, since it has about $n_{block}/2$ times total connections as the others. LD2 has MBD $1 + \log \frac{L}{n_{block}}$.

supervision (Lee et al., 2015) for the early layers helps the convergence of the network, even though the original DenseNet does not see performance impact from deep supervision. For simplicity, we place the auxiliary predictions at the end of each block. Let $x_i$ be a feature layer at the end of a block. Then the auxiliary prediction at $x_i$ takes as input $x_i$ along with $x_i$'s input features. Following (Hu et al., 2017), we put half of the total weighting in the final prediction and spread the other half evenly. After convergence, we take one extra epoch of training optimizing only the final prediction. We found this results in the lower validation error rate than always optimizing the final loss alone.

## 4 EXPERIMENTS

For visual recognition, we experiment on CIFAR10, CIFAR100 (Krizhevsky & Hinton, 2009), SVHN (Netzer et al., 2011), and ILSVRC2012 (Russakovsky et al., 2015).[1] We follow (He et al., 2016; Huang et al., 2017) for the training procedure and parameter choices. Specifically, we optimize using stochastic gradient descent with a moment of 0.9 and a batch size of 64 on CIFAR and SVHN. The learning rate starts at 0.1 and is divided by 10 after 1/2 and 3/4 of the total iterations are done. We train 250 epochs on CIFAR, 60 on SVHN, and 90 on ILSVRC. For CIFAR and SVHN, we specify a network by a pair $(n, g)$, where $n$ is the number of dense layers in each of the three dense blocks, and $g$, the growth rate, is the number of channels in each new layer.

### 4.1 IT MATTERS WHERE SHORTCUT CONNECTIONS ARE

This section verifies that short MBD is an important design principle by comparing the proposed Log-DenseNet V1 against two other intuitive connection strategies that also connects each layer $i$ to $1 + \log(i)$ previous layers. The first strategy, called **NEAREST** connects layer $i$ to its previous $\log(i)$ depths, i.e., $x_i = f_i(\text{concat}(\{x_{i-k} : k = 1, ..., \lfloor \log_b(i) \rfloor\}) ; \theta_i)$. The second strategy, called **EVENLY-SPACED** connects layer $i$ to $log(i)$ previous depths that are evenly spaced; i.e., $x_i = f_i(\text{concat}(\{x_{\lfloor i-1-k\delta \rceil} : \delta = \frac{i}{\log(i)} \text{ and } k = 0, 1, 2, ... \text{ and } k\delta \leq i - 1\}) ; \theta_i)$. Both methods above are intuitive. However, each of them has a MBD that is on the order of $O(\frac{L}{\log(L)})$, which is much higher than the $O(\log(L))$ MBD of the proposed Log-DenseNet V1. We experiment with networks whose $(n, g)$ are in $\{12, 32, 52\} \times \{16, 24, 32\}$, and show in Table 1 that Log-DenseNet almost always outperforms the other two strategies. Furthermore, the average relative increase of top-1 error rate using NEAREST and EVENLY-SPACED from using Log-DenseNet is $12.2\%$ and $8.5\%$, which is significant: for instance, (52,32) achieves 23.10% error rate using EVENLY-SPACED, which is about 10% relatively worse than the 20.58% from (52,32) using Log-DenseNet, but (52,16) using Log-DenseNet already has 23.45% error rate using a quarter of the computation of (52,32).

We also showcase the advantage of small MBD when each layer $x_i$ is connects to $\approx \frac{i}{2}$ number of previous layers. With these $O(L^2)$ total connections, NEAREST has a MBD of $\log L$, because

---

[1]CIFAR10 and CIFAR100 have 10 and 100 classes, and each have 50,000 training and 10,000 testing 32x32 color images. We adopt the standard augmentation to randomly flip left to right and crop 28x28 for training. SVHN contains around 600,000 training and around 26,000 testing 32x32 color images of numeric digits from the Google Street Views. We adopt the same pad-and-crop augmentations, and also apply Gaussian blurs. ILSVRC consists of 1.2 million training and 50,000 validation images from 1000 classes. We apply the same data augmentation for training as (He et al., 2016; Huang et al., 2017), and we report validation-set error rate from a single-crop of size 224x224 at test time.

| (n,g) | N, E GFLOPS | N+LD GFLOPS | D GFLOPS | CIFAR10 | | | | CIFAR100 | | | |
|---|---|---|---|---|---|---|---|---|---|---|---|
| | | | | N | E | N+LD | D | N | E | N+LD | D |
| (12,16) | 0.49 | 0.58 | 0.86 | 9.45 | 6.42 | 5.77 | 4.95 | 35.97 | 29.61 | 25.49 | 23.54 |
| (12,24) | 1.09 | 1.29 | 1.92 | 6.49 | 5.18 | 5.12 | 4.29 | 29.11 | 24.61 | 22.87 | 21.73 |
| (12,32) | 1.93 | 2.29 | 3.41 | 5.01 | 4.84 | 4.70 | 4.05 | 25.04 | 23.70 | 21.96 | 20.56 |
| (32,12) | 1.12 | 1.31 | 2.44 | 7.16 | 4.90 | 4.80 | 4.00 | 33.64 | 24.03 | 22.70 | 20.35 |
| (32,24) | 4.45 | 5.21 | 9.74 | 4.69 | 4.06 | 4.36 | 4.00 | 24.58 | 21.00 | 21.27 | 19.54 |
| (32,32) | 7.91 | 9.25 | 17.30 | 4.30 | 4.24 | 4.03 | 3.64 | 22.84 | 21.23 | 21.81 | 19.41 |
| (52,16) | 4.34 | 4.98 | 10.38 | 5.68 | 4.72 | 4.34 | 3.59 | 29.38 | 21.73 | 20.68 | 19.83 |

Table 2: Performance of connection patterns with $O(L^2)$ total connections. NEAREST (N), EVENLY-SPACED (E), and NearestHalfAndLog (N+LD) connect layer $i$ to about $i/2$ previous layers. DenseNet without block compression (D) connects $i$ to all previous $i - 1$ layers, and is thus about twice as expensive as the other three options. We highlight that N+LD greatly improves over N, because the few $\log L$ additional connections greatly reduced the MBD. The MBD of N, E, N+LD, and D are $\log L$, 2, 2, and 1.

| Method | GFLOPS | # Params (M) | Building | Tree | Sky | Car | Sign | Road | Pedestrian | Fence | Pole | Sidewalk | Cyclist | Mean IoU | Accuracy |
|---|---|---|---|---|---|---|---|---|---|---|---|---|---|---|---|
| SegNet[1] | - | 29.5 | 68.7 | 52.0 | 87.0 | 58.5 | 13.4 | 86.2 | 25.3 | 17.9 | 16.0 | 60.5 | 24.8 | 46.4 | 62.5 |
| FCN8[2] | - | 134.5 | 77.8 | 71.0 | 88.7 | 76.1 | 32.7 | 91.2 | 41.7 | 24.4 | 19.9 | 72.7 | 31.0 | 57.0 | 88.0 |
| DeepLab-LFOV[3] | - | 37.3 | 81.5 | 74.6 | 89.0 | 82.2 | 42.3 | 92.2 | 48.4 | 27.2 | 14.3 | 75.4 | 50.1 | 61.6 | na |
| Dilation8 + FSO[4] | - | 140.8 | **84.0** | **77.2** | 91.3 | **85.6** | **49.9** | 92.5 | 59.1 | 37.6 | 16.9 | 76.0 | **57.2** | 66.1 | 88.3 |
| FC-DenseNet67 (g=16)[5] | 40.9 | 3.5 | 80.2 | 75.4 | **93.0** | 78.2 | 40.9 | **94.7** | 58.4 | 30.7 | **38.4** | 81.9 | 52.1 | 65.8 | 90.8 |
| FC-DenseNet103 (g=16)[5] | 39.4 | 9.4 | 83.0 | **77.3** | **93.0** | 77.3 | 43.9 | 94.5 | **59.6** | 37.1 | 37.8 | **82.2** | 50.5 | 66.9 | 91.5 |
| LogDensenetV1-103 (g=24) | 42.0 | 4.7 | 81.6 | 75.5 | 92.3 | 81.9 | 44.4 | 92.6 | 58.3 | **42.3** | 37.2 | 77.5 | 56.6 | **67.3** | 90.7 |

Table 3: Performance on the CamVid semantic segmentation data-set. The column GFLOPS reports the computation on a 224x224 image in 1e9 FLOPS. We compare against 1 (Badrinarayanan et al., 2015), 2 (Long et al., 2015), 3 (Chen et al., 2016), 4 (Kundu et al., 2016), and 5 (Jégou et al., 2017).

we can halve $i$ (assuming $i > j$) until $j > i/2$ so that $i$ and $j$ are directly connected. EVENLY-SPACED has a MBD of 2, because each $x_i$ takes input from every other previous layer. Table 2 shows that EVENLY-SPACED significantly outperform NEAREST on CIFAR10 and CIFAR100, validating the importance of MBD. We also show that NEAREST can be greatly improved with just a few additional shortcuts to reduce the MBD. In the NEAREST scheme, $x_i$ already connects to $x_{i-1}, x_{i-2}, ..., x_{i/2}$. We then also connects $x_i$ to $x_{\lfloor i/4 \rfloor}, x_{\lfloor i/8 \rfloor}, x_{\lfloor i/16 \rfloor}, ....$ We call this scheme NearestHalfAndLog, and it has a MBD of 2, because any $j < i$ is either directly connected to $i$, if $j > i/2$, or $j$ is connected to some $i/\lfloor i/2^k \rfloor$ for some $k$, which is connected to $i$ directly. Fig. 1d illustrates the connections of this scheme. We observe in Table 2 that with this few $\log_i -1$ additional connections to the existing $\lceil i/2 \rceil$ ones, we drastically reduce the error rates to the level of EVENLY-SPACED, which has the same MBD of 2. These comparisons support our design principle: with the same number of connections at each depth $i$, the connection patterns with low MBD outperform the ones with high MBD.

## 4.2 LOG-DENSENET FOR TRAINING SEMANTIC SEGMENTATION FROM SCRATCH

Semantic segmentation assigns every pixel of input images with a label class, and it is an important step for understanding image scenes for robotics such as autonomous driving. The state-of-the-art training procedure (Zhao et al., 2017; Chen et al., 2016) typically requires training a fully-convolutional network (FCN) (Long et al., 2015) and starting with a recognition network that is trained on large data-sets such as ILSVRC or COCO, because training FCNs from scratch is prone to overfitting and is difficult to converge. Jégou et al. (2017) shows that DenseNets are promising for enabling FCNs to be trained from scratch. In fact, fully convolutional DenseNets (FC-DenseNets) are shown to be able to achieve the state-of-the-art predictions training from scratch without additional data on CamVid (Brostow et al., 2008) and GATech (Raza et al., 2013). However, the drawbacks of DenseNet are already manifested in applications on even relatively small images (360x480 resolution from CamVid). In particular, to fit FC-DenseNet into memory and to run it in reasonable speed, Jégou et al. (2017) proposes to cut many mid-connections: during upsampling, each layer is only directly connected to layers in its current block and its immediately previous block. Such connection strategy is similar to the NEAREST strategy in Sec. 4.1, which has already been shown to be less effective than the proposed Log-DenseNet in classification tasks. We now experimentally show that fully-convolutional Log-DenseNet (FC-Log-DenseNet) outperforms FC-DenseNet.

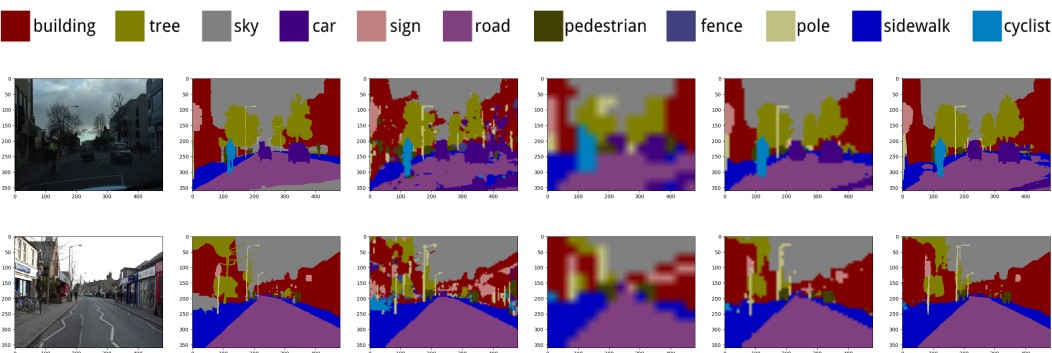

Figure 2: Each row: input image, ground truth labeling, and any scene parsing results at 1/4, 1/2, 3/4 and the final layer. Noting that the first half of the network downsamples feature maps, and the second half upsamples, we have the lowest resolution of predictions at 1/2, so that its prediction appear blurred.

**FC-Log-DenseNet 103.** Following (Jégou et al., 2017), we form FC-Log-DenseNet V1-103 with 11 Log-DenseNet V1 blocks, where the number of feature layers in the blocks are $4, 5, 7, 10, 12, 15, 12, 10, 7, 5, 4$. After each of the first five blocks, there is a transition that transforms and downsamples previous layers independently. After each of the next five blocks, there is a transition that applies a transposed convolution to upsample each previous layer. Both down and up sampling are only done when needed, so that if a layer is not used directly in the future, no transition is applied to it. Each feature layer takes input using the Log-DenseNet connection strategy. Since Log-DenseNet connections are sparse to early layers, which contain important high resolution features for high resolution semantic segmentation, we add feature layer $x_4$, which is the last layer of the first block, to the input set of all subsequent layers. This adds only one extra connection for each layer after the first block, so the overall complexity remains roughly the same. We do not form any other skip connections, since Log-DenseNet already provides sparse connections to past layers. We do not form FC networks using Log-DenseNet V2, because there are 11 blocks, so that V2 would multiply the final block cost by about 10. This is significant, because the final block already costs about half of the total FLOPS. We breakdown the FLOPS by blocks in the appendix Fig. 5b.

**Training details.** Our training procedure and parameters follow from those of FC-DenseNet (Jégou et al., 2017), except that we set the growth rate to 24 instead of 16, in order to have around the same computational cost as FC-DenseNet. We defer the details to the appendix. However, we also found auxiliary predictions at the end of each dense block reduce overfitting and produce interesting progression of the predictions, as shown in Fig. 2. Specifically, these auxiliary predictions produces semantic segmentation at the scale of their features using 1x1 conv layers. The inputs of the predictions and the weighting of the losses are the same as in classification, as specified in Sec. 3.3.

**Performance analysis.** We note that the final two blocks of FC-DenseNet and FC-Log-DenseNet cost half of their total computation. This is because the final blocks have fine resolutions, which also make the full DenseNet connection in the final two blocks prohibitively expensive. This is also why FC-DenseNets (Jégou et al., 2017) have to forgo all the mid-depth the shortcut connections in its upsampling blocks. Table 3 lists the Intersection-over-Union ratios (IoUs) of the scene parsing results. FC-Log-DenseNet achieves 67.3% mean IoUs, which is slightly higher than the 66.9% of FC-DenseNet. Among the 11 classes, FC-Log-DenseNet performs similarly to FC-DenseNet. Hence FC-Log-DenseNet achieves the same level of performance as FC-DenseNet with 50% fewer parameters and similar computations in FLOPS. This supports our hypothesis that we should minimize MBD when we have can only have a limited number of skip connections. FC-Log-DenseNet can potentially be improved if we reuse the shortcut connections in the final block to reduce the number of upsamplings.

### 4.3 Computational Efficiency of Sparse and Dense Networks

This section studies the trade-off between computational cost and the accuracy of networks on visual recognition. In particular, we address the question of whether sparser networks like Log-DenseNet perform better than DenseNet using the same computation. DenseNets can be very deep for image classification, because they have low resolution in the final block. In particular, a skip connection to

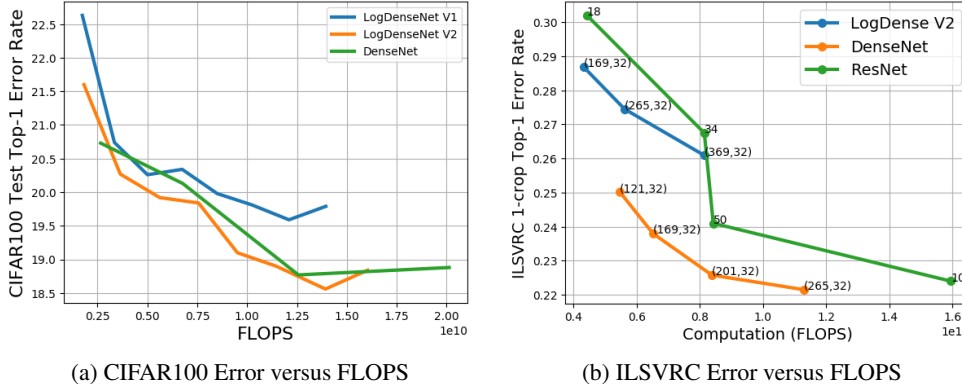

(a) CIFAR100 Error versus FLOPS      (b) ILSVRC Error versus FLOPS

Figure 3: **(a)** Using the same FLOPS, Log-DenseNet V2 achieves about the same prediction accuracy as DenseNets on CIFAR100. The DenseNets have block compression and are trained with dropouts. **(b)** On ILSVRC2012, Log-DenseNet 169, 265 have the same block sizes as DenseNets169, 265. Log-DenseNet369 has block sizes $8, 16, 80, 80$.

the final block costs $1/64$ of one to the first block. Fig. 3a illustrates the error rates on CIFAR100 of Log-DenseNet V1 and V2 and DenseNet. The Log-DenseNet variants have $g = 32$, and $n = 12, 22, 32, ..., 82$. DenseNets have $g = 32$, and $n = 12, 22, 32, 42$. Log-DenseNet V2 has around the same performance as DenseNet on CIFAR100. This is partially explained by the fact that most pairs of $x_i, x_j$ in Log-DenseNet V2 are cross-block, so that they have the same MBD as in Densenets thanks to the compressed early blocks. The within block distance is bounded by the logarithm of the block size, which is smaller than 7 here. Log-DenseNet V1 has similar error rates as the other two, but is slightly worse, an expected result, because unlike V2, backpropagation distances between a pair $x_i, x_j$ in V1 is always $\log |i - j|$, so on average V1 has a higher MBD than V2 does. The performance gap between Log-DenseNet V1 and DenseNet also gradually widens with the depth of the network, possibly because the MBD of Log-DenseNet has a logarithmic growth. We observe similar effects on CIFAR10 and SVHN, whose performance versus computational cost plots are deferred to the appendix. These comparisons suggest that to reach the same accuracy, the sparse Log-DenseNet costs about the same computation as the DenseNet, but is capable of scaling to much higher depths. We also note that using naïve implementations, and a fixed batch size of 16 per GPU, DenseNets (52, 24) already have difficulties fitting in the 11GB RAM, but Log-DenseNet can fit models with $n > 100$ with the same $g$. We defer the plots for number of parameters versus error rates to the appendix as they look almost the same as plots for FLOPS versus error rates.

On the more challenging ILSVRC2012 (Russakovsky et al., 2015), we observe that Log-DenseNet V2 can achieve comparable error rates to DenseNet. Specifically, Log-DenseNet V2 is more computationally efficient than ResNet (He et al., 2016) that do not use bottlenecks (ResNet18 and ResNet34): Log-DenseNet V2 can achieve lower prediction errors with the same computational cost. However, Log-DenseNet V2 is not as computationally efficient as ResNet with bottlenecks (ResNet 50 and ResNet101), or DenseNet. This implies there may be a trade-off between the shortcut connection density and the computation efficiency. For problems where shallow networks with dense connections can learn good predictors, there may be no need to scale to very deep networks with sparse connections. However, the proposed Log-DenseNet provides a reasonable trade-off between accuracy and scalability for tasks that require deep networks, as in Sec. 4.2.

## 5 CONCLUSIONS AND DISCUSSIONS

We show that short backpropagation distances are important for networks that have shortcut connections: if each layer has a fixed number of shortcut inputs, they should be placed to minimize MBD. Based on this principle, we design Log-DenseNet, which uses $O(L \log L)$ total shortcut connections on a depth-$L$ network to achieve $1 + \log L$ MBD. We show that Log-DenseNets improve the performance and scalability of *tabula rasa* fully convolutional DenseNets on CamVid. Log-DenseNets also achieve competitive results in visual recognition data-sets, offering a trade-off between accuracy and network depth. Our work provides insights for future network designs, especially those that cannot afford full dense shortcut connections and need high depths, like FCNs.

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

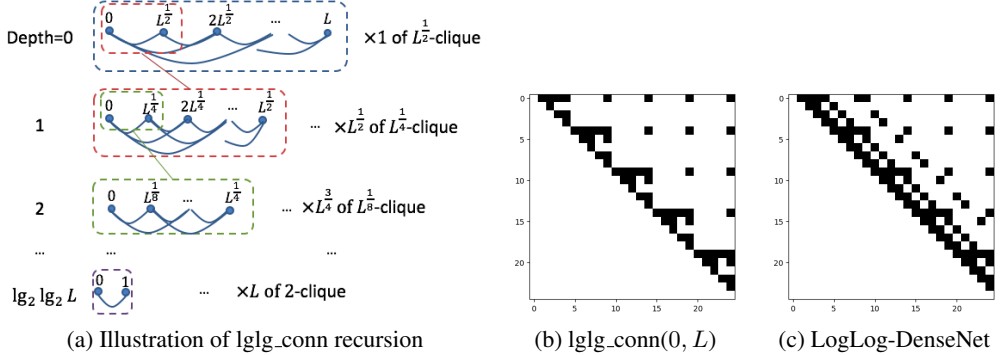

(a) Illustration of lglg_conn recursion      (b) lglg_conn$(0, L)$      (c) LogLog-DenseNet

Figure 4: **(a)**The tree of recursive calls of `lglg_conn`. **(b)** LogLog-DenseNet augments each $x_i$ of `lglg_conn`$(0, L)$ with Log-DenseNet connections until $x_i$ has at least four inputs.

## Appendix

## A    LOGLOG-DENSENET

Following the short MBD design principle, we further propose LogLog-DenseNet, which uses $O(L \log \log L)$ connections to achieve $1 + \log \log L$ MBD. For the clarity of construction, we assume there is a single block for now, i.e., $n_{block} = 1$. We add connections in LogLog-DenseNet recursively, after we initialize each depth $i = 1, ..., L$ to take input from $i - 1$, where layer $i = 0$ is the initial convolution that transform the image to a $2g$-channel feature map. Fig. 4 illustrates the recursive calls. Formally, the recursive connection-adding function is called `lglg_conn`$(s, t)$, where the inputs $s$ and $t$ represent the start and the end indices of a segment of contiguous layers in $0, ..., L$. For instance, the root of the recursive call is `lglg_conn`$(0, L)$, where $(0, L)$ represents the segment of all the layers $0, ..., L$. `lglg_conn`$(s, t)$ exits immediately if $t - s \leq 1$. If otherwise, we let $\delta = \lfloor \sqrt{t - s + 1} \rfloor$, $K = \{s\} \cup \{t - k\delta : t - k\delta \geq s \text{ and } k = 0, 1, 2, ..., \}$, and let $a_1, ..., a_{|K|}$ be the sorted elements of $K$. **(a)** Then we add dense connections among layers whose indices are in $K$, i.e., if $i, j \in K$ and $i > j$, then we add $x_j$ to the input set of $x_i$. **(b)** Next for each $k = 1, ..., |K| - 1$, we add $a_k$ to the input set of $x_j$ for each $j = a_k + 1, ..., a_{k+1}$. **(c)** Finally, we form $|K| - 1$ number of recursive `lglg_conn` calls, whose inputs are $(s_k, s_{k+1})$ for each $k = 1, ..., |K| - 1$.

If $n_{block} > 1$, we reuse Log-DenseNet V1 transition to scale each layer independently, so that when we add $x_j$ to the input set of $x_i$, the appropriately scaled version of $x_j$ is used instead of the original $x_j$. We defer to the appendix the formal analysis on the recursion tree rooting at `lglg_conn`$(0, L)$, which forms the connection in LogLog-DenseNet, and summarize the result as follows.

**Proposition A.1.** *LogLog-DenseNet of $L$ feature layers has at most $1.5L \log \log L + o(L \log \log L)$ connections, and a MBD at most $\log \log L + n_{block} + 1$.*

Hence, if we ignore the independent transitions and think them as part of each $x_i$ computation, the MBD between any two layers $x_i, x_j$ in LogLog-DenseNet is at most $2 + \log \log L$, which effectively equals 5, because $\log \log L < 3.5$ for $L < 2545$. Furthermore, such short MBD is very cheap: on average, each layer takes input from 3 to 4 layers for $L < 1700$, which we verify in B.2. We also note that without step (b) in the `lglg_conn`, the MBD is $2 + 2 \log \log L$ instead of $2 + \log \log L$.

**Bottlenecks.** In DenseNet, since each layer takes input from the concatenation of all previous layers, it is necessary to have bottleneck structures (He et al., 2016; Huang et al., 2017), which uses 1x1 conv to shrink the channel size to $4g$ first, before using 3x3 conv to generate the $g$ channel of features in each $x_i$. In Log-DenseNet and LogLog-DenseNet, however, the number of input layers is so small that bottlenecks actually increase the computation, e.g., most of LogLog-DenseNet layers do not even have $4g$ input channels. However, we found that bottlenecks are cost effective for increasing the network depth and accuracy. Hence, to reduce the variation of structures, we use bottlenecks and fix the bottleneck width to be $4g$. For LogLog-DenseNet, we also add Log-DenseNet connections from nearest to farthest for each $x_i$ until either $x_i$ has four inputs, or there are no available layers. For $L < 1700$, this increases average input sizes only to $4.5 \sim 5$, which we detail in the next section.

# B   FORMAL ANALYSIS OF LOG-DENSENET AND LOGLOG-DENSENET

## B.1   PROOF OF PROPOSITION 3.1

*Proof.* We call BD $(x_i, x_j)$ the back-propagation distance from $x_i, x_j$, which is the distance between the two nodes $x_i, x_j$ on the graph constructed for defining MBD in Sec. 3.2. The scaling transition happens only once for each scale during backpropagation; i.e., there are at most $n_{block} - 1$ number of transitions between any two layers $x_i, x_j$.

Since the transition between each two scales happens at most once, in between two layers $x_i, x_j$, we first consider $n_{block} = 1$, and add $n_{block} - 1$ to the final distance bound to account for multiple blocks.

We now prove the proposition for $n_{block} = 1$ by induction on $|i - j|$. Without loss of generality we assume $i > j$. The base case: for all $i > j$ such that $i = j + 1$, we have BD $(x_i, x_j) = 1$. Now we assume the induction hypothesis that for some $t \geq 0$ and $t \in \mathbb{N}$, such that for all $i > j$ and $i - j \leq 2^t$, we have BD $(x_i, x_j) \leq t + 1$. Then for any two layers $i > j$ such that $i - j \leq 2^{t+1}$, if $i - j \leq 2^t$, then by the induction hypothesis, BD $(x_i, x_j) \leq t+1 < t+2$. If $2^{t+1} \geq i - j > 2^t$, then we have $k := i - 2^t > j$, and $k = i - 2^t \leq 2^{t+1} - 2^t + j = j + 2^t$. So that $k - j \in (0, 2^t]$. Next by the induction hypothesis, BD $(x_k, x_j) \leq t + 1$. Furthermore, by the connections of Log-DenseNet, $x_i$ takes input directly from $x_{i-2^t}$, so that BD $(x_i, x_k) = 1$. Hence, by the triangle inequality of distances in graphs, we have BD $(x_i, x_j) \leq$ BD $(x_i, x_k)$ + BD $(x_k, x_j) \leq 1 + (t + 1)$. This proves the induction hypothesis for $t + 1$, so that the proposition follows, i.e., for any $i \neq j$, BD $(x_i, x_j) \leq \log |i - j| + 1$.

$\square$

## B.2   PROOF OF PROPOSITION A.1

*Proof.* Since the transition between each two scales happens at most once, in between two layers $x_i, x_j$, we again first consider $n_{block} = 1$, and add $n_{block} - 1$ to the final distance bound to account for multiple blocks.

(**Number of connections.**)  We first analyze the recursion tree of `lglg_conn(0, L)`. In each `lglg_conn(s, t)` call, let $n = t - s + 1$ be the number of layers on the segment $(s, t)$. Then the interval of key locations $\delta = \lfloor \sqrt{t - s + 1} \rfloor = \lfloor \sqrt{n} \rfloor$, and the key location set $K = \{s\} \cup \{t - k\delta : t - k\delta \geq s$ and $k = 0, 1, 2, ..., \}$ has a cardinality of $|K| = 1 + \lceil \frac{n-1}{\lfloor \sqrt{n} \rfloor} \rceil \in (\sqrt{n}, 2.5 + \sqrt{n})$. Hence the step (a) of `lglg_conn(s, t)` in Sec. A adds $1 + 2 + 3 + ... + (|K| - 1) = 0.5|K|(|K| - 1)$. Step (b) of `lglg_conn(s, t)` then creates $(n - |K|)$ new connections, since the ones among $x_s, ..., x_t$ that are not given new connections are exactly $x_i$ in $K$. Hence, `lglg_conn(s, t)` using step (a),(b) increases the total connections by

$$c(n) = n + 0.5|K|^2 - 1.5|K| < 1.5n + \sqrt{n} + 3.125. \tag{3}$$

Step (c) instantiate $|K| - 1 \leq \sqrt{n} + 1.5$ calls of `lglg_conn`, each of which has an input segment of length at most $\delta \leq \sqrt{n} + 1$. Hence, let C(n) be the number of connections made by the recursive call `lglg_conn(s, t)` for $n = t - s + 1$, then we have the recursion

$$C(n) \leq (\sqrt{n} + 1.5)C(\sqrt{n} + 1) + c(n). \tag{4}$$

Hence, the input segment length takes a square root in each depth until the base case at length 2. The depth of the recursion tree of `lglg_conn(s, t)` is then $1 + \log \log(t - s + 1)$. Furthermore, the connections made on each depth $i$ of the tree is $1.5n + o(n)$, because at each depth $i = 0, 1...$, $c(n^{2^{-i}} + o(n^{2^{-i}})) = 1.5n^{2^{-i}} + o(n^{2^{-i}})$ connectons are made in each `lglg_conn`, and the number of calls is 1 for $i = 0$, and $\Pi_{j=0}^{i}(n^{2^{-j}} + 1.5) = n^{1-2^{-i}} + o(n^{1-2^{-i}})$. Hence the total connections in LogLog-DenseNet is $C(L + 1) = 1.5L \log \log L + o(L \log \log L)$.

(**Back-propagation distance.**)   First, each $x_i$ for $i \in [s, t]$ in the key location set $K$ of `lglg_conn(s, t)` is in the input set of $x_t$. Second, for every $x_i$, and for every call `lglg_conn(s, t)` in the recursion tree such that $s < i \leq t$ and $(s, t) \neq (0, L)$, we know step (b) adds $x_s$ to the input set of $x_i$. Hence, we can form a back-propagation path from any $x_i$ to $x_j$ $(i > j)$ by first using a connection from step (b) to go to a key location of the `lglg_conn(s, t)` call such that $[s, t]$ is the

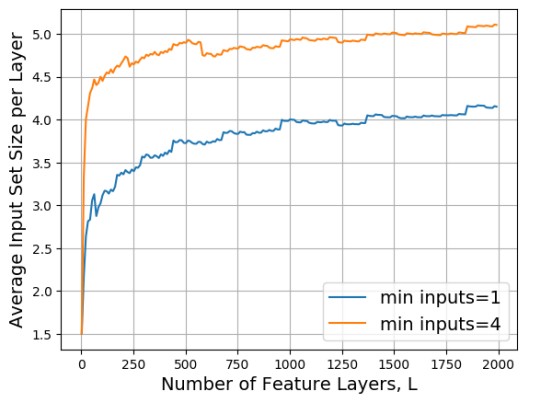 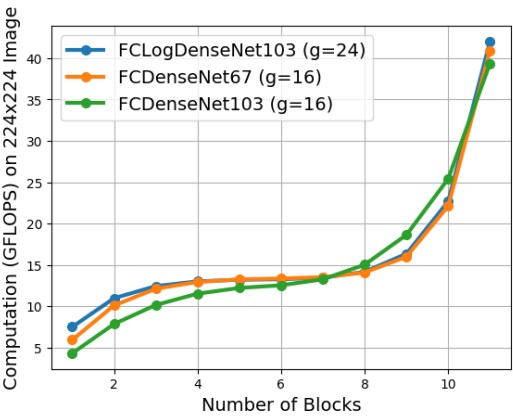

(a) Average number of inputs per Layer in LogLog-DenseNet

(b) Blocks versus Computational Cost in FCNs

Figure 5: **(a)** In `lglg_conn(0, L)`, i.e., min inputs = 1, each layer on average takes input from 3 to 4 layers. If we force input size to be four when possible using Log-DenseNet connection pattern, i.e., min inputs = 4, we increase the average input size by 1 to 1.5. **(b)** Computational cost (in FLOPS) distribution through the 11 blocks in FC-DenseNet and FC-Log-DenseNet. Half of the computations are from the final two blocks due to the high final resolutions. We compute the FLOPS assuming the input is a single 224x224 image.

smallest interval in the recursion tree such that $i, j \in [s, t]$. Then we can continue the path to $x_j$ by following the recursion calls whose input segments include $j$ until $j$ is in a key location set. The longest path is then the depth of the recursion tree plus one initial jump, i.e., $2 + \log \log L$.

$\square$

### B.3 LogLog-DenseNet Layers on Average Has Five Connections in Practice

Figure 5a shows the average number of input layers for each feature layer in LogLog-DenseNet. Without augmentations, `lglg_conn` on average has 3 to 4 connections per layer. With augmentations using Log-DenseNet, we desire each layer to have four inputs if possible. On average, this increases the number of inputs by 1 to 1.5 for $L \in (10, 2000)$.

## C Additional Experimental Results

### C.1 CamVid Training Details

We follow Jégou et al. (2017) to optimize the network using 224x224 random cropped images with RMSprop. The learning rate is 1e-3 with a decay rate 0.995 for 700 epochs. We then fine-tune on full images with a learning rate of 5e-4 with the same decay for 300 epochs. The batch size is set to 6 during training and 2 during fine-tuning. We train on two GTX 1080 GPUs. We use no pre-processing of the data, except left-right random flipping. Following Badrinarayanan et al. (2015), we use the median class weighting to balance the weights of classes, i.e., the weight of each class $C$ is the median of the class probabilities divided by the over the probability of $C$.

### C.2 Computational Efficiency on CIFAR10 and SVHN

Fig. 6a and Fig. 6b illustrate the trade-off between computation and accuracy of Log-DenseNet and DenseNets on CIFAR10 and SVHN. Log-DenseNets V2 and DenseNets have similar performances on these data-sets: on CIFAR10, the error rate difference at each budget is less than 0.2% out of 3.6% total error; on SVHN, the error rate difference is less than 0.05% out of 1.5%. Hence, in both cases, the error rates between Log-DenseNet V2 and DenseNets are around 5%.

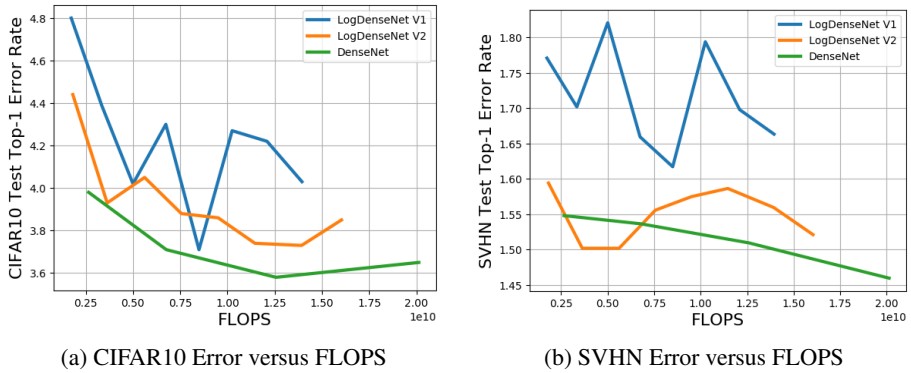

(a) CIFAR10 Error versus FLOPS  (b) SVHN Error versus FLOPS

Figure 6: On CIFAR10 and SVHN, Log-DenseNet V2 and DenseNets have very close error rates ($< 5\%$ relatively difference) at each budget.

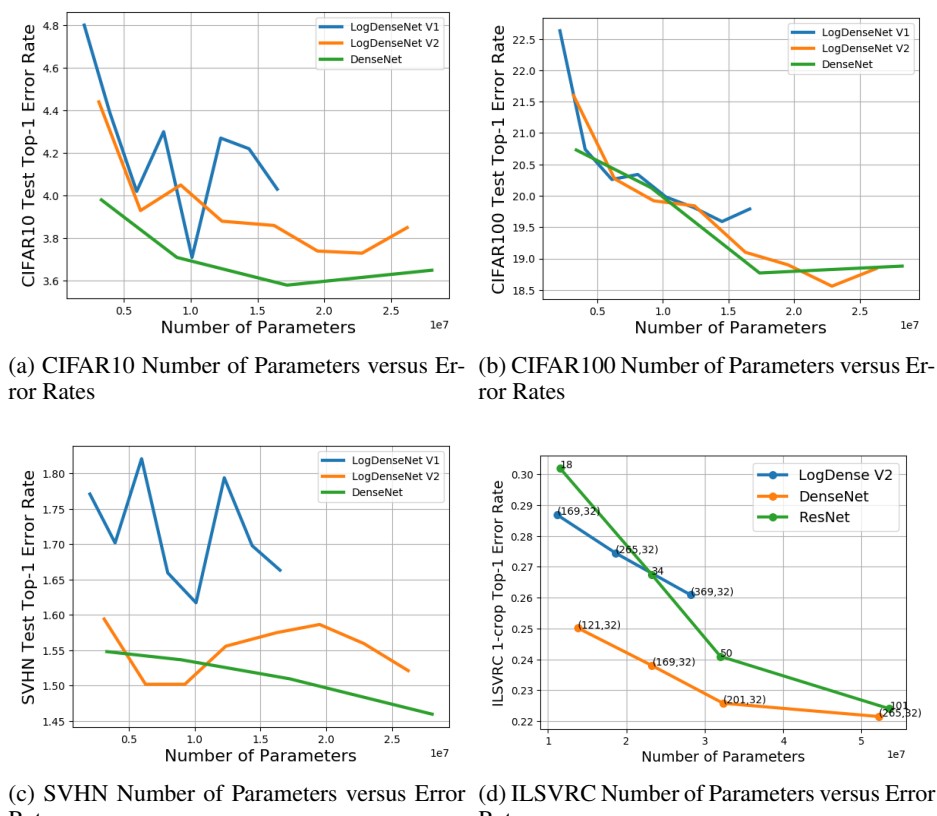

(a) CIFAR10 Number of Parameters versus Error Rates  (b) CIFAR100 Number of Parameters versus Error Rates

(c) SVHN Number of Parameters versus Error Rates  (d) ILSVRC Number of Parameters versus Error Rates

Figure 7: The number of parameter used in the naïve implementation versus the error rates on various data-sets.

## C.3 NUMBER OF PARAMETER VERSUS ERROR RATES.

Figure 7 plots the number of parameters used by Log-DenseNet V2, DenseNet, and ResNet versus the error rates on the image classification data-sets, CIFAR10, CIFAR100, SVHN, ILSVRC. We assume that DenseNet and Log-DenseNet use naïve implementations.

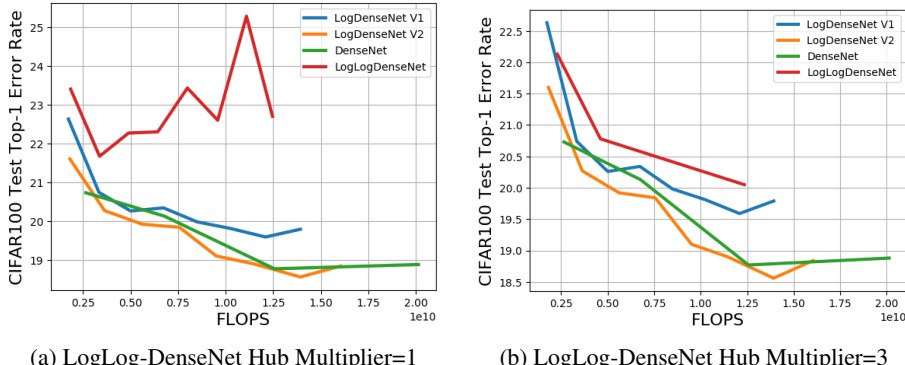

(a) LogLog-DenseNet Hub Multiplier=1          (b) LogLog-DenseNet Hub Multiplier=3

Figure 8: Performance of LogLog-DenseNet (red) with different hub multiplier (1 and 3). Larger hubs allow more information to be passed by the hub layers, so the predictions are more accurate.

## C.4  LogLog-DenseNet Experiments and More Principles than MBD

This section experiments with LogLog-DenseNet and show that there are more that just MBD that affects the performance of networks. Ideally, since LogLog-DenseNet have very small MBD, its performance should be very close to DenseNet, if MBD is the sole decider of the performance of networks. However, we observe in Fig. 8a that LogLog-DenseNet is not only much worse than Log-DenseNet and DenseNet in terms accuracy at each given computational cost (in FLOPS), it is also widening the performance gap to the extent that the test error rate actually increases with the depth of the network. This suggests there are more factors at play than just MBD, and in deep LogLog-DenseNet, these factors inhibit the networks from converging well.

One key difference between LogLog-DenseNet's connection pattern to Log-DenseNet's is that the layers are not symmetric, in the sense that layers have drastically different shortcut connection inputs. In particular, while the average input connections per layer is five (as shown in Fig. 5a), some nodes, such as the nodes that are multiples of $L^{\frac{1}{2}}$, have very large in-degrees and out-degrees (i.e., the number of input and output connections). These nodes are given the same number of channels as any other nodes, which means there must be some information loss passing through such "hub" layers, which we define as layers that are densely connected on the depth zero of `lglg_conn` call. Hence a natural remedy is to increase the channel size of the hub nodes. In fact, Fig. 8b shows that by giving the hub layers three times as many channels, we greatly improve the performance of LogLog-DenseNet to the level of Log-DenseNet. This experiment also suggests that the layers in networks with shortcut connections should ensure that high degree layers have enough capacity (channels) to support the amount of information passing.

## C.5  Additional Semantic Segmentation Results

We show additional semantic segmentation results in Figure 9. We also note in Figure 5b how the computation is distributed through the 11 blocks in FC-DenseNets and FC-Log-DenseNets. In particular, more than half of the computation is from the final two blocks because the final blocks have high resolutions, making them exponentially more expensive than layers in the mid depths and final layers of image classification networks.

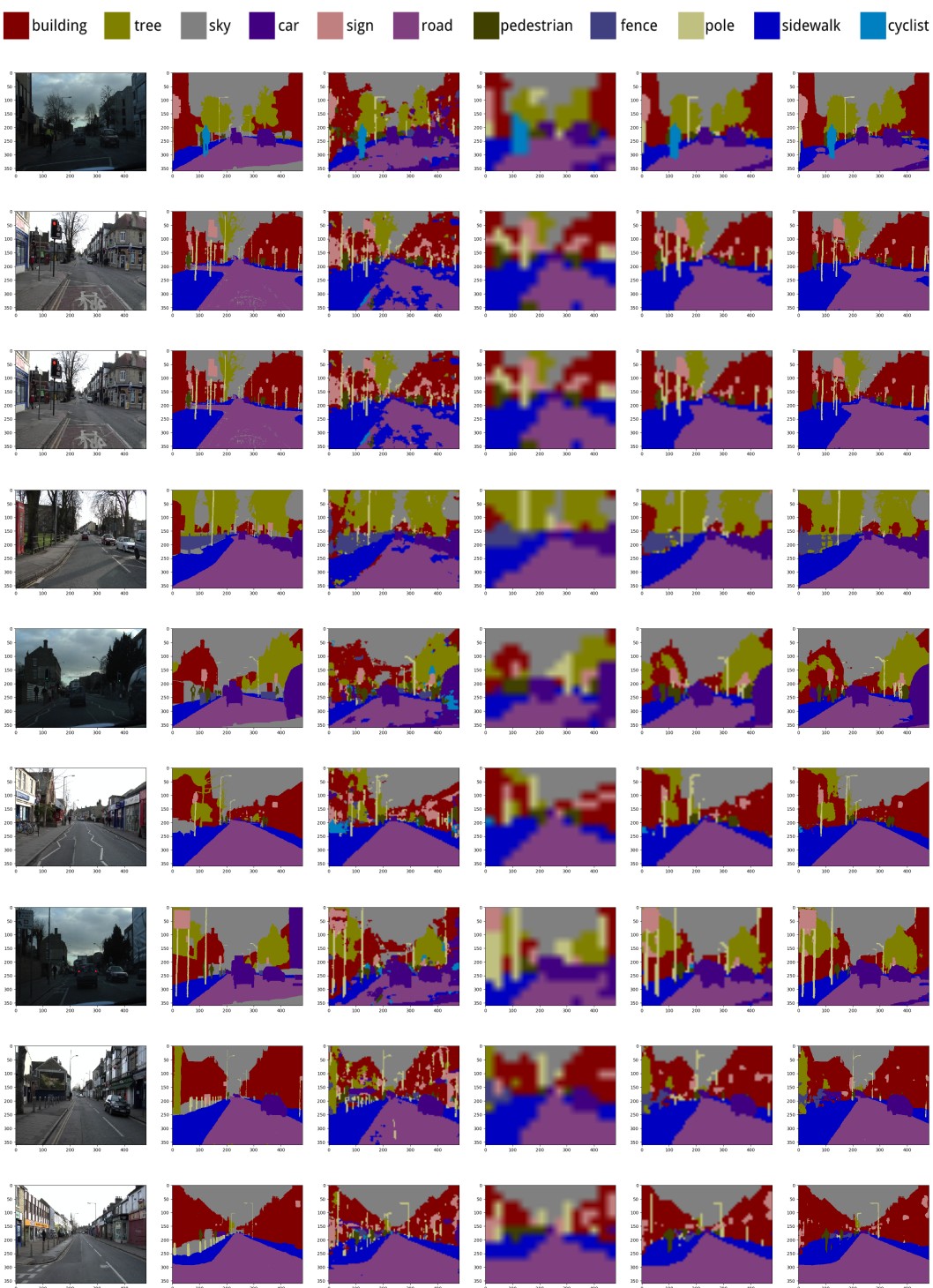

Figure 9: Each row: input image, ground truth labeling, and any scene parsing results at 1/4, 1/2, 3/4 and the final layer.

