# OpenReview forum: "Log-DenseNet: How to Sparsify a DenseNet"
_ICLR.cc/2018/Conference — Reject_

### Official Review · AnonReviewer2 · 2017-11-26

**Rating:** 6
**Confidence:** 4

**Review:**

This paper investigates how to impose layer-wise connections in DenseNets most efficiently. The authors propose a connection-pattern, which connects layer i to layer i-2^k, k=0,1,2... The authors also propose maximum backpropgation distance (MBD) for measuring the fluency of gradient flow in the network, and justify the Log-DenseNet's advantage in this framework. Empirically, the author demonstrates the effectiveness of Log-DenseNet by comparing it with two other intuitive connection patterns on CIFAR datasets. Log-DenseNet also improves on FC-DenseNet, where the connection budget is the bottleneck because the feature maps are of high resolutions.


Strengths:
1. Generally, DenseNet is memory-hungry if the connection is dense, and it is worth studying how to sparsify a DenseNet. By showing the improvements on FC-DenseNet, Log-DenseNet demonstrates good potential on tasks which require upsampling of feature maps.
2. The ablation experiments are well-designed and the visualizations of connectivity pattern are clear.

Weakness:
1. Adding a comparison with Log-DenseNet and vanilla DenseNet in the Table 2 experiment would make the paper stronger. Also, the NearestHalfAndLog pattern is not used in any latter visual recognition experiments, so I think it's better to just compare LogDenseNet with the two baselines instead. Despite there are CIFAR experiments on Log-DenseNet in latter sections, including results here would be easier to follow.
2. I would like to see the a comparison with the DenseNet-BC in the segmentation and CIFAR classification tasks, which uses 1x1 conv layers to reduce the number of channels. It should be interesting to study whether it is possible to further sparsify DenseNet-BC, as it has much higher efficiency.
3. The improvement of efficiency on classifications task is not that significant.

---

### Official Review · AnonReviewer1 · 2017-11-27
**Interesting idea to sparsify skip connections in DenseNets, well executed experiments**

**Rating:** 6
**Confidence:** 4

**Review:**

This paper introduces a new connectivity pattern for DenseNets, which encourages short distances among layers during backpropagation and gracefully scales to wider and deeper architectures. Experiments are performed to analyze the importance of the skip connections’ place in the context of image classification. Then, results are reported for both image classification and semantic segmentation tasks.

The clarity of the presentation could be improved. The main contribution of the paper is a network design that places skip connections to minimize the distances between layers, increasing the distance from 1 to 1 + log L when compared to traditional DenseNets. This design principle allows to mitigate the memory required to train DenseNets, which is critical for applications such as semantic segmentation where the input resolution has to be recovered.

Experiments seem well executed; the authors consider several sparse connectivity patterns for DenseNets and provide empirical evidence highlighting the advantages of having a short maximum backpropagation distance (MBD). Moreover, they provide an analysis on the trade-off between the performance of a network and its computational cost.

Although literature review is quite extensive, [a] might be relevant to discuss in the Network Compression section.
[a] https://arxiv.org/pdf/1412.6550.pdf

It is not clear why Log-DenseNets would be easier to implement than DenseNets, as mentioned in the abstract. Could the authors clarify that?

In Tables 1-2-3, it would be good to add the results for Log-DenseNet V2. Adding the MBD of each model in the tables would also be beneficial.

In Table 3, what does “nan” accuracy mean? (DeepLab-LFOV)

Finally, the authors might want to review the citep/cite use in the manuscript.

---

### Official Review · AnonReviewer3 · 2017-11-27
**Nice idea, the presentation could be improved.**

**Rating:** 5
**Confidence:** 4

**Review:**

The paper proposes a nice idea of sparsification of skip connections in DenseNets. The authors decide to use a principle for sparsification that would minimize the distance among layers during the backpropagation.

The presentation of the paper could be improved. The paper presents an elegant and simple idea in a dense and complex way making the paper difficult to follow. E. g., Fig 1 d is discussed in Appendix and not in the main body of the paper, thus, it could be moved to Appendix section.

Table 1 and 3 presents the results only for LogDenseNet V1, would it be possible to add results for V2 that have different MBD. Also, the budget for the skip connections is defined as log(i) in Table 1 and Table 2 has the budget of log(i/2), would it be possible to add the total number of skip connections to the tables? It would be interesting to compare the total number of skip connections in Jegou et. al. to LogDenseNet V1 in Table 3.

Other issues:
- Table 3, has an accuracy of nan. What does it mean? Not available or not a number?
- L is used as the depth, however, in table 1 it appears as short for Log-DenseNetV1. Would it be possible to use another letter here?
- “…, we make x_i also take the input from x_{i/4}, x_{i/8}, x_{i/16}…”. Shouldn’t x_{1/2} be used too?
- I’m not sure I understand the reasons behind blurred image in Fig 2 at ½. It is mentioned that “it and its feature are at low resolution”. Could the authors comment on that?
- Abstract: “… Log-DenseNets are easier than DenseNet to implement and to scale.” It is not clear why would LogDenseNets be easier to implement.

---

### Decision · Program_Chairs · 2018-01-29
**ICLR 2018 Conference Acceptance Decision**

**Decision:**

Reject

**Comment:**

The paper presents an empirical study into sparse connectivity patterns for DenseNets.

Whilst sparse connectivity is potentially interesting, the paper does not make a strong argument for such sparse connectivity patterns: in particular, the results on ImageNet suggest that sparse connectivity performs substantially worse than full connectivity (at the same FLOPS-level, Log-DenseNet obtains ~2.5% lower accuracy than baseline DenseNet models, and the best Log-DenseNet is ~4% worse than the best DenseNet). On CamVid, both network architectures appear to perform on par.

The paper motivates the model architecture by the high memory consumption of DenseNets but, frankly, that is a very weak motivation: DenseNets are actually very memory-efficient if implemented correctly (https://arxiv.org/pdf/1707.06990.pdf). The fact that such implementations are not well-supported by TensorFlow/PyTorch is a shortcoming of those deep-learning frameworks, not in DenseNets. (In fact, the memory management features that deep-learning frameworks have implemented to make residual networks memory-efficient (for instance, caching GPU memory allocation in PyTorch) are far more complex than the "thousand lines of C++" currently needed to implement a DenseNet correctly.) Such issues will likely be resolved relatively soon by better implementations, and are hardly a good motivation for a different network architecture.